# Effect of Nano-Zinc Oxide, Rice Straw Compost, and Gypsum on Wheat (*Triticum aestivum* L.) Yield and Soil Quality in Saline–Sodic Soil

**DOI:** 10.3390/nano14171450

**Published:** 2024-09-05

**Authors:** Mahmoud El-Sharkawy, Modhi O. Alotaibi, Jian Li, Esawy Mahmoud, Adel M. Ghoneim, Mohamed S. Ramadan, Mahmoud Shabana

**Affiliations:** 1School of the Environment and Safety Engineering, School of Emergency Management, Jiangsu University, Zhenjiang 212013, China; mahmoud.elsharkawy@agr.tanta.edu.eg (M.E.-S.); jianli@ujs.edu.cn (J.L.); 2Soil and Water Department, Faculty of Agriculture, Tanta University, Tanta P.O. Box 31527, Egypt; 3Department of Biology, College of Science, Princess Nourah bint Abdulrahman University, P.O. Box 84428, Riyadh 11671, Saudi Arabia; 4Field Crops Research Institute, Agricultural Research Center, Giza P.O. Box 12511, Egypt; adelrrtc.ghoneim@gmail.com; 5Soils, Water and Environment Research Institute (SWERI), Agricultural Research Center, Giza P.O. Box 12619, Egypt; msaramadan86@gmail.com (M.S.R.); shabanamma@gmail.com (M.S.)

**Keywords:** specific surface area, numbers of root galls, microbial biomass carbon, soil amendments, soil health

## Abstract

The salinity and alkalinity of soils are two fundamental factors that limit plant growth and productivity. For that reason, a field study conducted at Sakha Agric. Res. Station in Egypt during the 2022–2023 winter season aimed to assess the impact of gypsum (G), compost (C), and zinc foliar application in two images, traditional (Z_1_ as ZnSO_4_) and nanoform (Z_2_ as N-ZnO), on alleviating the saline–sodic conditions of the soil and its impact on wheat productivity. The results showed that the combination of gypsum, compost, and N-ZnO foliar spray (G + C + Z_2_) decreased the soil electrical conductivity (EC), sodium adsorption ratio (SAR), and exchangeable sodium percentage (ESP) by 14.81%, 40.60%, and 35.10%, respectively. Additionally, compared to the control, the G + C + Z_2_ treatment showed improved nutrient content and uptake as well as superior wheat biomass parameters, such as the highest grain yield (7.07 Mg ha^−1^), plant height (98.0 cm), 1000-grain weight (57.03 g), and straw yield (9.93 Mg ha^−1^). Interestingly, foliar application of N-ZnO was more effective than ZnSO_4_ in promoting wheat productivity. Principal component analysis highlighted a negative correlation between increased grain yield and the soil EC and SAR, whereas the soil organic matter (OM), infiltration rate (IR), and plant nutrient content were found to be positively correlated. Furthermore, employing the k-nearest neighbors technique, it was predicted that the wheat grain yield would rise to 7.25 t ha^−1^ under certain soil parameters, such as EC (5.54 dS m^−1^), ESP (10.02%), OM (1.41%), bulk density (1.30 g cm^−3^), infiltration rate (1.15 cm h^−1^), and SAR (7.80%). These results demonstrate how adding compost and gypsum to foliar N-ZnO can improve the soil quality, increase the wheat yield, and improve the nutrient uptake, all of which can support sustainable agriculture.

## 1. Introduction

Wheat (*Triticum aestivum* L.) is an indispensable cereal crop in Egypt, covering approximately 1.32 million hectares of land and boasting an annual production of 8.45 million tons. Its importance is underscored by its pivotal role in the Egyptian diet, with consumption reaching 20.1 million tons in 2018/19, reflecting a 1.5% increase every year, according to a USDA report cited by Abdalla et al. [1]. Wheat is the main crop in Egypt and is considered the best dietary food for the population of Egypt due to its high carbohydrate and protein content [2]. In addition, wheat straw is used as a fodder for animals [3]. However, wheat cultivation faces formidable challenges, particularly from abiotic stresses, which have garnered global attention. Among these stresses, salinity poses a formidable threat to wheat production, particularly in arid and semi-arid regions, as highlighted by Godoy et al. [4] and Hossain et al. [5].

Soil degradation resulting from salinity has become an imperious global concern, particularly under the population projections indicating a surge to 9.6 billion by 2050, while cultivated soils face an annual decline of 1–2%, as noted by Hossain [6]. Based on varying assessments, there are an estimated 831 to 932 million hectares of salinized soils globally, mostly in arid and semi-arid regions [7]; roughly 23% of these soils are saline and 37% are sodic [8]. The severity of the issue is further highlighted by projections from Liu et al. [9], indicating that 50% of cultivated lands will succumb to salinity by 2050. In Egypt, approximately 0.9 million hectares of irrigated areas suffer from salinity as a result of irrigation from agricultural drainage or high ground water levels [10]. The dual challenge of saline–sodic soils encompasses issues stemming from both salinity and structural deterioration [11]. High salinity inhibits plant growth through various mechanisms, including ion toxicity, osmotic pressure-induced water deficit, and nutritional imbalances [12]. As for sodicity, it causes the dispersion of clay in the soil as a result of the presence of a large amount of Na^+^ in the soil solution or exchange sites, in addition to the deterioration of its biological and chemical properties [10]. Given the pivotal role of agriculture in Egypt’s economy, reclaiming salt-affected soils emerges as a crucial strategy to expand cultivable land and mitigate the adverse impacts of soil degradation.

Reclaiming saline–sodic soils necessitates a multifaceted approach aimed at replacing the exchangeable sodium phase with soluble divalent cations in the soil solution, thereby enhancing soil flocculation [13]. Subsequently, the leaching of soluble salts from the soil profile becomes imperative [14]. Various amendments have been identified as effective strategies for ameliorating salt-affected soils. A commonly practiced method involves the application of organic and chemical conditioners to enhance the soil quality [15,16]. Research indicates that the incorporation of organic substances such as crop straw and biochar can significantly enhance the physicochemical characteristics of soil, thereby improving the overall soil quality [17,18]. Similarly, gypsum application, as a traditional practice, has been found to ameliorate soil flocculation conditions by reducing the exchangeable sodium percentage [18,19]. The integration of compost and gypsum has emerged as a particularly effective soil amendment, leading to reductions in the soil pH, salinity, and sodium adsorption ratio (SAR), while also enhancing the bulk density and hydraulic conductivity [20,21]. Moreover, organic amendments have been found to exert a positive influence on the physical properties of saline–sodic soils by expediting cation exchange on the soil surface and facilitating salt leaching from the rhizosphere area [22,23]. This comprehensive approach underscores the importance of integrating various amendments to effectively reclaim and improve the quality of saline–sodic soils.

Mani et al. [24] enumerate the effects of nanoparticles as very good adsorbents in the soil that could dominate nutrient and pollutant conveyance, organizing the stabilization of organic matter, and stimulating the formation of new mineral phases. Nutrients released at the nano scale possess characteristics that can be tailored to the specific needs of crops, providing relief on demand and controlling the release of chemical fertilizers, thereby regulating plant growth and enhancing target efficiency [25,26]. Compared to conventional fertilizers, nanofertilizers are poised to revolutionize crop cultivation by improving growth, productivity, and nutrient uptake, while also reducing losses and mitigating environmental impacts [27]. This transformative potential underscores the significance of ongoing research and application of nanotechnology in agriculture, offering a sustainable pathway towards enhancing agricultural productivity and sustainability.

Zinc (Zn) is an important micronutrient that is necessary for plant growth and is crucial for the metabolism of both proteins and carbohydrates [28]. It is necessary in small amounts, yet it is necessary for proper plant development [29]. The efficiency of Zn depends on its absorption and translocation within the plants [30]. The conventional fertilizers for Zn are EDTA-Zn chelate or Zn sulphate (ZnSO_4_), which could be applied in both foliar and ground amendments [31] and have been reported to exhibit low efficiency [32]. Subsequently, in response to the limitations of conventional fertilizers, nanofertilizers present a promising solution for the controlled release of target nutrients, addressing issues such as soil contamination, low agronomic productivity, and abiotic stresses in plants [33,34,35]. Dimkpa et al. [36] showed that nanoparticles are more reactive than their bulk counterparts because they contain a significant surface area and the ability to produce unique features that improve nutrient transport. Research conducted using N-ZnO on *Pisum sativum* has demonstrated notable enhancements in production, biomass, root development, germination rates, and chlorophyll concentration compared to conventional ZnSO_4_ [37]. Nanofertilizers in foliar systems have been found to be suitable for field application because they feed nutrients to plants in a more regulated manner than salt fertilizers in addition to their role in mitigating toxicity caused by replenishing the soil with the same nutrients [38]. While N-ZnO’s effects on plant physiological responses and soil pollutants are well documented, its impact on the physicochemical properties of saline–sodic soil remains relatively unexplored. This research aims to study the effect of combining soil amendments such as compost, gypsum, and foliar spraying with zinc in its traditional and nanoforms on the soil quality and productivity of wheat plants growing in saline–sodic soil. Through this research, we seek to deepen our understanding of how these interventions can contribute to sustainable agriculture practices amidst challenges posed by soil degradation and nutrient deficiency.

## 2. Materials and Methods

### 2.1. Experimental Location and Design

In the North Delta area of Egypt, at the Sakha Agricultural Research Station Farm in the Kafr El-Sheikh Governorate, a field trial was carried out in the winter of 2022–2023. The aim was to examine the impact of various soil amendments (gypsum (G), compost (C), and a combination of gypsum and compost), along with Zn foliar use in the forms of ZnSO_4_ (Z_1_) and N-ZnO (Z_2_), at rates of 0 and 2 g L^−1^, and their interactions on soil chemical properties and wheat productivity growing in saline–sodic soil. The site is located at 30°56′53′′ E longitude and 31°05′36′′ N latitude with an elevation of about 6 m above sea level in arid climate conditions. The maximum, minimum, and average air temperature is 20.47, 13.18, and 16.83 °C, average precipitation of 14.73 mm, and average humidity of 38% during the entire experiment. The physicochemical characteristics are presented in Table 1.

The experimental field was laid out and divided into 36 plots, each of size 2 m × 2 m. The trial used a randomized complete block design, with twelve separate treatments that were duplicated three times. The treatments included a varied range of interventions:

Control (CK): with no amendments; gypsum (G): as G requirements; rice straw compost (C); foliar spraying of ZnSO_4_ (Z_1_); foliar N-ZnO (Z_2_); gypsum plus compost (G + C); gypsum plus foliar ZnSO_4_ (G + Z_1_); gypsum + foliar nano-Zn (G + Z_2_); compost + foliar ZnSO_4_ (C + Z_1_); compost + foliar nano-Zn (C + Z_2_); gypsum + compost + foliar ZnSO_4_ (G + C + Z_1_); and gypsum + compost + foliar nano-Zn (G + C + Z_2_) as in the following block diagram:



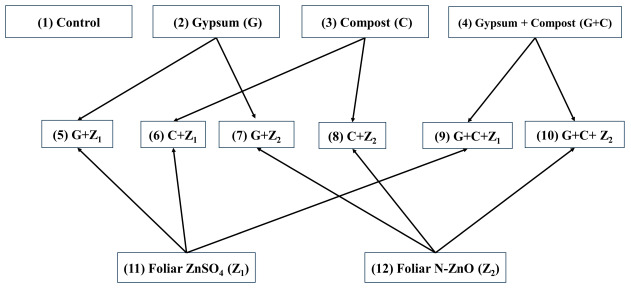



Gypsum was used at an amount of 9.55 Mg ha^−1^ in accordance with G requirements (GR). Rice straw compost, recommended by Sarwar et al. [39], was applied at an amount 12 Mg ha^−1^. Zinc sulfate and N-ZnO were administered as foliar applications at a concentration of 2 g L^−1^.

Rice straw compost and G were thoroughly incorporated into the surface soil depth (0–30 cm) before cultivation. The foliar sprays of Zn treatments were divided into two dosages: the first was administered after forty-five days of planting (December 2022), before the tilling stage, and the second after sixty days of planting (January 2023). The chemical constitution of the rice straw compost (C) is detailed in Table 2. Gypsum requirements were determined following the guidelines outlined by [40], aimed at decreasing the initial exchangeable sodium percentage (*ESP*) to 10% within the soil matrix of the surface depth (0–30 cm), as calculated by the provided equation:(1)$$GR=\left({ESP}_{i}-{ESP}_{f}\right)\times CEC\times 1.72\times 2.4$$
where *GR*: *G* requirement (t ha^−1^) in the 0–30 cm soil, *ESP_i_*: initial soil *ESP*, *ESP_f_*: the target *ESP* in the soil, Na^+^: exchangeable sodium content, and *CEC*: cation exchange capacity (cmol_c_ kg^−1^).

The grains of wheat (*Triticum aestivum* L.) genotype (*Gemmiza 11*) were sown at an amount of 144 kg ha^−1^ on 15 November 2022 and harvested at maturity stage on 7 April 2023. Phosphorus was supplied as super mono phosphate (18% P_2_O_5_) at a rate of 75 kg ha^−1^ during field preparation. Additionally, potassium sulfate (48% K_2_O) was added at 120 kg ha^−1^, divided into two applications: the first with the initial irrigation and the second before the tillering stage. In addition, nitrogen fertilizer in the form of urea (46% N) was added at an amount of 180 kg ha^−1^, split into two doses. The initial dose was administered after the first watering irrigation, with the second dose applied during the subsequent irrigation. All other agricultural practices were carried out in accordance with the recommendations provided by the Ministry of Agriculture for wheat cultivation in the North Delta Area.

### 2.2. N-ZnO Preparation and Characterization

According to Rajendran et al. [41], the zinc acetate precursor approach was used to create zinc oxide nanoparticles (N-ZnO) by the sol–gel process. Initially, separate solutions of sodium hydroxide (NaOH) and zinc acetate (1M) were prepared. After the zinc acetate solution dissolved completely, the NaOH solution (2M) was put in dropwise and stirred for eighteen h. This process resulted in the formation of white precipitation, which was subsequently filtered and heated in an oven (at 90 °C for 2 h) before being calcined in a muffle furnace at 400 °C.

The crystalline phase of N-ZnO was investigated using X-ray diffractometry (ARL EQUINOX 100-Thermofisher Scientific Inc., Miami, FL, USA), while the size and morphology of the particles were visualized through transmission electron microscopy (TEM) imaging techniques (JSM-7610F, JEOL Ltd., Peabody, MA, USA). The average size of N-ZnO particles was determined from Scherrer’s formula [42]:(2)D=Kλ/βCosϴ
where *D* represents the particle size, *K* is the Scherrer constant, *λ* denotes the X-ray beam wavelength (1.54, 184 Å), *β* signifies the FWHM of the peak, and *θ* represents the Bragg angle.

### 2.3. Soil Analysis

For every experimental unit, surface soil samples were taken at intervals of 0–30 cm. These samples were air-dried, crushed, sieved to pass through a 2.0 mm sieve, and homogenized. Soil pH was measured by a pH meter (model H12211-02, Thermofisher, HANNA, Boston, MA, USA) in soil paste, and electrical conductivity (EC) was measured using an EC meter (model CON2700, EUTECH, Vernon Hills, IL, USA). Particle size distribution was analyzed using the Pipette method described by Scheldrick [43], bulk density (*BD*) determined as described by Briggs [44], and subsequently total open porosity (TP) was calculated assuming soil particle density (*PD*) as 2.65 g cm^−3^ described by [45] as follows:(3)$$TP(\%)=1-\frac{BD}{PD}$$

Cation exchange capacity (CEC) analysis was conducted using the ammonium acetate method as outlined by Page et al. [46]. Soil organic matter was determined through wet digestion using a 1 N potassium dichromate solution and sulfuric acid, following the Walkley and Black method as outlined by [47]. For the available nitrogen (N), the Kjeldahl method was employed following the protocol of extraction with a 2 M KCl solution established by the Soil Survey Staff [48]. The samples of soils were analyzed spectrophotometrically using the atomic absorption chemical analysis method. Available phosphorus (P) content was determined spectrophotometrically using the ascorbic acid method after extraction with a 0.5 M sodium bicarbonate solution at pH 8.30, following the procedure outlined by Olsen [49]. The available potassium (K) content was determined using a flame photometer after extraction with 1.0 N ammonium acetate at pH 7, as described by Page et al. [46].

The calculation of the sodium adsorption ratio (SAR) followed the method outlined by Chi et al. [50], wherein the concentrations of cations are expressed in mmol L^−1^ as follows:(4)$$SAR(\%)=\frac{Na}{\sqrt{\frac{Ca+Mg}{2}}}$$

Subsequently, *ESP* was computed using the equation reported by Kim et al. [51] as follows:(5)ESP (%)=1.95+1.03SAR

At the end of the experiment, the removal sodium efficiency (*RSE*) of soils was calculated according to Amer [52] as
(6)$$RSE=\frac{{ESP}_{i}-{ESP}_{f}}{{ESP}_{i}}\times 100$$
where *ESP_i_* is the initial *ESP* and *ESP_f_* is the final *ESP* at the end of the experiment.

### 2.4. Plant Sampling and Analysis

Plant samples including straw and grain of wheat plants were harvested at the maturity stage (120 days after planting) in 1 m^2^ plots representing each treatment to assess various plant characteristics. These characteristics encompassed plant biomass, measured as grain yield (Mg ha^−1^), straw yield (Mg ha^−1^), plant height (cm), and 1000-grain weight (g). The harvest index was computed by dividing the dry matter of the grain yield by the total dry matter of both the grain and straw yield as reported by [53].

The harvested wheat straw and grain samples from each treatment were subjected to a series of preparatory steps. Initially, they were oven-dried at 70 °C and subsequently ground to facilitate further analysis. Digestion of the samples was carried out using a mixture of sulfuric and perchloric acids, following the method outlined by [54]. According to Cottenie et al. [55], a flame photometer was used to measure the total K content. N and P contents were analyzed according to [46], while total Zn was determined using the procedure as described by Liu et al. [56]. The nutrient uptake of nitrogen, phosphorus, and potassium by the grains and straw was calculated by multiplying the respective nutrient percentages by the dry grain weight per plant and expressing the results as kilograms per hectare (kg ha^−1^).

### 2.5. Statistical Analysis

The experiments were conducted in triplicate, and a one-way analysis of variance (ANOVA) with one factor was performed using IBM-SPSS Statistics (version 29). Replications were considered random and all other variables were considered fixed effects, and the significant levels were set to (5%). To compare the means, a Duncan multiple range test (DMRT) was used with a significance of *p* < 0.05. Principal component analysis (PCA) was conducted using Minitab 2022 LLC (version 21.4.0) to investigate the differences and relationship between different quality variables. The experimental architecture for ANN building is presented in Figure 1.

## 3. Results

### 3.1. TEM Image and XRD of Prepared N-ZnO

The morphological characteristics of nanoparticles are of significant interest, as morphology often influences various properties of nanoparticles. Illustrated in Figure 2 are transmission electron microscopy (TEM) images showcasing the morphology of nano-zinc oxide (N-ZnO) powder synthesized via a chemical method. These images provide insights into the diffusion and structural characteristics of N-ZnO particles, revealing ample void spaces conducive to accommodating volume expansion, ensuring good cycling performance and enhanced rate capacity [57]. Furthermore, the TEM images indicate that N-ZnO particles exhibit a non-circular morphology, resembling polygonal-like structures, with an average size estimated to be approximately 40–50 nm. Moezzi et al. [58] highlighted the several dimensions structure as the most favorable nanostructure compared to other one-dimensional nanostructures. X-ray diffraction (XRD) serves as a rapid analytical technique primarily utilized for the phase identification of crystalline materials, providing valuable insights into unit cell dimensions [59]. In Figure 3, the XRD pattern of the prepared N-ZnO powder reveals distinctive peaks, including major characteristic peaks at 100, 002, 101, 102, 110, and 100. These peaks correspond to the wurtzite or hexagonal quartzite structure of N-ZnO, as determined through comparison with standard [JCPDF] cards. Specifically, these peaks are observed at 28.18°, 31.8°, 34.34°, 37.58°, 40.46°, 66.32°, and 71.51°, respectively, in agreement with the findings of Sowri Babu et al. [60]. The average size of N-ZnO particles is estimated to be 31 nm, determined from the Full Width at Half Maximum (FWHM) equation applied to the most intense peak corresponding to the 101 planes, located at 34.34°.

### 3.2. Effect of Studied Soil Amendments on Some Soil Characteristics

The soil amendments effectively mitigated the soil salinity, as illustrated in Table 3. The data demonstrated that the application of gypsum at 100% GR in combination with compost at 12 Mg ha^−1^ and the N-ZnO foliar application (G + C + Z_2_) treatment recorded the lowest EC with a value of 5.52 dS m^−1^ after wheat harvesting. This was closely followed by the G + C + Z_1_ treatment, with an average EC value of 5.54 dS m^−1^, demonstrating improvements of 12.52% and 12.26%, respectively, compared to the control treatment. Further insights into the changes in electrical conductivity (EC) following the application of different treatments are presented in Figure 4. The G + C + Z_2_ treatment exhibited the most substantial impact, with a 14.81% reduction in salinity, followed by the G + C + Z_1_ treatment, which recorded a decrease of 14.66%. In contrast, the control treatment demonstrated the lowest change in ECe, with a value of 2.62%.

Table 3 presents the findings concerning the soil sodium adsorption ratio (SAR), exchangeable sodium percentage (ESP), and residual sodium carbonate (RSC). The data revealed that the integration of gypsum and compost resulted in mitigating the soil sodicity, showing a good behavior in ameliorating the records of the SAR, ESP, and RSE with improvement percentages of 32.68%, 23.66%, and 89.63%, respectively. Moreover, when combined with foliar zinc treatments, whether in conventional or nanoforms, the efficiency in mitigating the soil sodicity was further pronounced, with enhancement percentages reaching 40.60%, 35.10%, and 91.50% for the SAR, ESP, and RSE, respectively, compared to the control treatment.

Table 4 illustrated that the soil organic matter was significantly (*p* ≤ 0.05) affected by different soil amendments compared to the control (CK). All treatments that received compost alone or in combination with other additions increased the soil organic matter with a frequent descending order as follows: G + C + Z_2_ > G + C + Z_1_ > G + C > C + Z_2_ > C + Z_1_ > C with values of 1.43, 1.42, 1.42, 1.39, 1.39, and 1.37%, respectively. Table 4 also illustrates that the soil physical properties encompassing the bulk density, soil porosity, and infiltration rate were affected by compost, gypsum, and their combinations. The data illustrated that the foliar application of zinc fertilizers showed no differences in the soil physical characteristics in both forms with N-ZnO exploring slight enhancements, while the integration of both foliar Zn application with compost and/or gypsum investigated more consolidations in soil properties. For the bulk density, there were no significant differences (*p* ≤ 0.05) between all treatments except with treatments applied with the combination of compost and chemical amendments. The G + C + Z_2_, G + C + Z_1,_ and C + Z_2_ treatments recorded the lowest bulk densities with average values of 1.33, 1.34, and 1.35 g cm^−3^, respectively. Furthermore, the application of rice straw compost explored valuable enhancements in both the soil porosity and soil infiltration rate (IR), as presented in Table 4. The sole addition of rice straw compost increased the total porosity by 5% at the end of the experiment in comparison to the initial result, while the G + C + Z_2_ treatment showed more amelioration in the total porosity, with 8.1% more than the initial record. With the same trend, the infiltration rate increased with all traits treated, with rice straw compost recording the highest value with the G + C + Z_2_ treatment (1.12 mm h^−1^) in comparison to the control.

### 3.3. Effect of Studied Soil Amendments on Yield and Biomass of Wheat Plants

The effects of compost, gypsum, foliar zinc fertilizers, and their combinations on the wheat (*Triticum aestivum* L.) plant biomass are presented in Table 5. The data reveal that treatments incorporating N-ZnO resulted in a notable increase in the wheat plant biomass, including the grain yield, straw yield, 1000-grain weight, harvest index, and plant height, compared to amendments using conventional ZnSO_4_ either alone or combined with compost or gypsum. Moreover, the integration of compost and gypsum led to a substantial enhancement in the wheat plant biomass, ranging from 1.5 to 2 times more than the control treatment. Among these combinations, the G + C + Z_2_ treatment exhibited the most significant increase in the plant biomass, with the grain yield reaching 7.07 Mg ha^−1^, straw yield at 9.93 Mg ha^−1^, 1000-grain weight recorded as 57.03 Mg ha^−1^, and a plant height of 98.0 cm. These results underscore the enhanced efficiency of N-ZnO in promoting these variables compared to other treatments.

### 3.4. Effect of Studied Soil Amendments on Nutrient Content of Wheat Plants

The nutrient contents (N, P, K, and Zn) in both the grains and straw of the wheat plants are shown in Table 6. There were significant differences (*p* < 0.01) in the nutrient contents of the wheat plants affected by different soil and foliar amendments. It was noticed that the integration between inorganic amendments via the gypsum and zinc foliar application treatments resulted in reducing the grain and straw NPK and Zn contents of the wheat plants, while the incorporation between organic and inorganic additions via the compost and foliar Zn treatments resulted in increasing the grain and straw NPK and Zn contents of the wheat plants. On the other hand, the application of N-ZnO demonstrated superior enhancement in the wheat plant N, P, K, and Zn contents. The results indicated that the G + C + Z_2_ treatment increased the N, P, K, and Zn contents in the grains with percentages of 84.9%, 257.5%, 48.1%, and 53.1%, respectively, compared to the control. Moreover, the G + C + Z_2_ treatment elevated the P, K, and Zn contents in the wheat straw by 0.23%, 1.29%, and 4.81 mg kg^−1^, respectively, while the G + C + Z_1_ treatment showed the highest N content in the wheat straw at 0.84%, with no significant difference compared to the G + C + Z_2_ treatment. A glance at Figure 5 illustrates that the total N, P, K, and Zn uptakes by the wheat plants were affected significantly (*p* < 0.01) by gypsum, compost, and Zn foliar application either individually or combined as compared to the control. Meanwhile, the data revealed that the total uptakes of N, P, K (kg ha^−1^), and Zn (g ha^−1^) by the wheat plants clearly increased approximately according to the following descending order: G + C + Z_2_ ˃ G + C + Z_1_ ˃ G + C ˃ Z_2_ > C + Z_2_ > C + Z_1_ > G + Z_2_.

The principal component analysis (PCA) employed using different soil variables including (EC, ESP, IR, BD, and OM) in conjunction with the nutrition conditions of wheat grains comprising (N, P, and K) correlated to the grain yield is depicted in Figure 6. The chart illustrates that the first two components explained 78.6% of the correlation, as shown in Figure 6A. The observations in Figure 6B illustrate that the EC and ESP negatively affected both components with the wheat grain yield, while the soil organic matter and infiltration rate were highly positively affected in the wheat yield. Furthermore, the grain yield appeared to closely associate with the plant nutrients via N, P, and K.

## 4. Discussion

### 4.1. Soil Characteristics

N-ZnO proved its capabilities in ameliorating saline–sodic soil properties. The results demonstrate the effectiveness of soil amendments in mitigating soil salinity, which is crucial for optimizing agricultural productivity in saline-affected areas. Gypsum, compost, and N-ZnO foliar application emerged as promising strategies in reducing the soil salinity, as evidenced by the significant improvements in the electrical conductivity (EC) values. The combination of gypsum at 100% gypsum requirement (GR) with organic amendments and N-ZnO foliar application (G + C + Z_2_) exhibited the most pronounced reduction in salinity. These results are in line with [61,62]. As for sodicity, a possible explanation for the decrease in the ESP, SAR, and RSC with the combination of gypsum and compost in Table 3 is a result of the improvement in the soil porosity or may be due to decreasing Na^+^ or increasing Ca^+2^ [63]. Several studies reported that the combined organic and inorganic amendments to sodic soil resulted in reducing the SAR and therefore ESP of soil [39,64,65]. Gypsum and compost amendments contribute to soil quality improvement by enhancing the soil structure and reducing sodium levels, while foliar zinc treatments provide additional benefits by promoting plant health and nutrient uptake, which indirectly influence soil sodicity [63,66]. Shaaban et al. [67] explained that during organic matter decomposition, CO_2_ is increased and large spontaneous amounts of H^+^ are released, which enhance the dissolution of CaCO_3_ and unleash more Ca^+2^ that exchanges with Na^+^ ions on soil colloids. Furthermore, in our results, the application of zinc treatments either in ZnSO_4_ or N-ZnO form fulfilled the improvement in the effect of all treatments especially the combined gypsum and compost treatments. This may be due to the acidity effect of compost and gypsum of soil, which led to more mobilization of some thrifty soluble Zn into the available form [68], which in turn may interact with different anion compounds such as CO_3_^−2^, HCO_3_^−^, SO_4_^−2^, or Cl^−,^ resulting in decreasing sodium cations by accelerating its leaching with irrigation water, as described by Tejada et al. [69]. The data demonstrated that the combination of N-ZnO with either compost or gypsum or their combination resulted in ameliorating the soil salinity and sodicity. This could be attributed to several factors. N-ZnO nanoparticles have a higher surface area to volume ratio compared to traditional ZnSO_4_, allowing for better nutrient absorption by plant roots [70]. This enhanced nutrient uptake can lead to improved plant health and growth, which indirectly contributes to soil salinity and sodicity management. N-ZnO particles possess high reactivity due to their small size and large surface area, allowing for better interaction with soil ions. When combined with gypsum and compost, N-ZnO can facilitate ion exchange processes, leading to reduced sodium accumulation and improved soil sodicity management [71].

The increase in SOM in all treatments, whether they received C alone or in conjunction with other amendments, is shown in Table 4. Two explanations exist for the increase in SOM in wheat plants receiving compost additions after harvesting: firstly, the proliferation of different bacteria in these byproducts, and secondly, its effect on root zone protections and plant growth [72]. Compost, being rich in organic matter, introduces beneficial microorganisms and organic compounds into the soil. These microorganisms help decompose organic residues in the compost, releasing nutrients and forming stable organic matter compounds in the soil [73]. Gypsum, on the other hand, addresses soil sodicity by improving the soil physical properties and decreasing the Na^+^ content [74]. When C and G are combined, their synergistic effects can further enhance the SOM in saline–sodic soil. Table 4 shows that the integration of both foliar Zn applications with C and/or G investigated more consolidations in the soil physical properties. Bulk density is an indicator of soil compaction. According to Cui et al. [75], the presence of Na^+^ causes compaction in sodic soils. When G is applied, either by itself or in combination with C, exchangeable Na^+^ ions are replaced with Ca^+2^ ions. This decreases clay dispersion, lowers the bulk density, and increases the soil porosity [76]. N-ZnO, being readily absorbed by plants, can enhance plant growth and root development [77]. As plants grow, their roots can penetrate and loosen the soil, increasing the soil pore spaces and thus reducing the bulk density [78]. Additionally, improved plant growth resulting from N-ZnO application can lead to increased organic matter input to the soil through root exudates and decaying root material, further improving the soil structure and reducing the bulk density [79]. Additionally, the increased plant biomass resulting from foliar N-ZnO application combined with compost and gypsum (G + C + Z_2_) can enhance the soil infiltration rate. More extensive root systems and improved soil structure facilitate water movement through the soil profile, reducing surface runoff and increasing water infiltration rates [78].

### 4.2. Wheat Plant Characteristics

The beneficial function of prior amendments in salt-affected soils can be related to boosting plant tolerance to salinity during physiological growth stages and improving some soil properties. Similar findings were achieved by [39,62,80], who demonstrated that the combination of C and G had a good impact on plant development under saline environments. Day et al. [81] hypothesized that the combination of C and chemical conditioners would enhance the effectiveness of chemical amendments by mobilizing soil surface sodium, which in turn enhances the soil chemical and physical characteristics, which is reflected in plant production, as well as the availability of nutrients during C decomposition [82].

Concerning the effect of foliar application, the N-ZnO foliar application displayed an overall improvement in the plant biomass than conventional ZnSO_4_. N-ZnO significantly (*p* < 0.01) increased the wheat growth and yield parameters, as presented in Table 5. Watson et al. [83] investigated the role of N-ZnO in comparison to ZnSO_4_ in the soil and found that the addition of N-ZnO to the alkaline soil increased the root zone of wheat plants, and added that the combination with organic amendments showed an adsorption efficiency of N-ZnO and affected its solubility from nanoparticles. Hussein et al. [84] stated that the favorable effects of spraying of N-ZnO on plant biomass were attributable to increased nutrient usage efficiency, improved plant physiological activity, and reduced soil environmental dangers caused by soil toxicity and contamination. The G + C + Z_2_ treatment in Table 5 showed significant enhancements in the grain yield, reaching 83.63%, straw yield at 27.86%, harvest index at 76.35%, 1000-grain weight recorded as 25.54%, and a plant height of 54.74% compared to the control. Several studies have reported significant improvements in plant biomass with the application of foliar nano-ZnO combined with compost and gypsum [85,86]. This enhancement in plant productivity indicates improved yield potential, which is crucial for meeting the demands of growing populations and ensuring food security. The small size and high surface area of the nanoparticles (surface area of nano-ZnO in this study = 73.10 m^2^ g^−1^) enable them to penetrate the leaf cuticle and epidermis, allowing for rapid absorption by the plant [87]. This direct foliar uptake ensures prompt availability of zinc for crucial physiological processes, such as enzyme activation, photosynthesis, and hormone regulation, which are essential for optimal plant growth and productivity [88,89]. Furthermore, nano-ZnO nanoparticles possess inherent bioactive properties that can stimulate various biochemical and physiological processes within the plant. These nanoparticles can act as signaling molecules, eliciting stress-responsive pathways and enhancing the ability of the plant to withstand environmental stresses such as drought and salinity [90,91]. Additionally, foliar nano-ZnO has been reported to enhance photosynthetic efficiency and chlorophyll synthesis in plants [92]. By optimizing the chloroplast structure and function, nano-ZnO nanoparticles facilitate greater light absorption and utilization, leading to increased photosynthetic rates and carbon assimilation. This, in turn, results in improved biomass accumulation, higher yields, and better overall plant productivity.

Table 6 demonstrates a significant rise in the nutrient contents (N, P, K, and Zn) in the grains and straw of wheat plants when treated with (G + C + Z_2_). According to Yang et al. [93], there is a correlation between organic amendments and gypsum that boosts biological activity by releasing certain physiological precursors and amino acids. This could explain the observed increase in root growth.

According to Verma et al. [94], applying farm manure in conjunction with mineral fertilizer can raise the levels of accessible potassium, phosphorus, and humus as well as nitrogen. According to Hussein et al. [84], cotton plants grown in saline soil had higher NPK concentrations when N-ZnO was treated topically. When Burman et al. [95] examined the effects of N-ZnO and ZnSO_4_ on chickpea, they found that N-ZnO enhanced the nutritional components of the plant and that this was because it reduced reactive oxygen species (ROS), which slowed superoxide dismutase and lipid peroxidation.

The total N, P, K, and Zn uptakes by the wheat plants were affected significantly (*p* < 0.01) by treatments individually or combined as compared to the control. Nano-zinc foliar application increased the total nutrient uptakes by wheat (*Triticum aestivum*) plants as compared to ZnSO_4_ foliar application and the control. These may be due to the vital physiological roles in plant cells, which promote the uptake of plant nutrients. Nanofertilizers are anticipated to enhance the crop uptake of micronutrients and decrease soil macronutrient losses, as demonstrated by Liu et al. [27]. Zinc foliar application gave a significant effect on the concentrations of macro/micronutrients in the grain and straw yield of wheat as compared with the control treatment. The principal component analysis (PCA) in Figure 6 illustrates that the EC and ESP negatively affected both components with the wheat grain yield producing plants with fewer tillers and lighter grains, as reported by Eynard et al. [96]. Meanwhile, the OM and IR were positively correlated with the enhancement in the wheat yield. Soil organic matter plays a crucial role in the soil structure and function, influencing various physical, chemical, and biological properties that are essential for plant growth and productivity by enhancing aggregation, porosity, and water retention capacity [97]. Increased levels of OM promote the formation of stable soil aggregates, which create macropores (>30 µm) and micropores (7–30 µm) within the soil matrix [98]. The literature revealed that each one level of microaggregates more than 55% led to an increase in the contacts between the solids in the macroaggregates by 10 times. These pores allow for better air and water movement through the soil profile, facilitating improved infiltration rates. Additionally, soil organic matter enhances soil moisture retention capacity, reducing the risk of water stress and drought-induced yield losses [99]. Furthermore, the grain yield was closely associated with the N, P, and K nutrients in wheat plants as these nutrients play critical roles in various physiological processes essential for plant growth, development, and ultimately grain formation [100].

Applying a supervised machine learning approach, specifically the k-nearest neighbors (KNN) algorithm with Model Baverage, offers a robust method for estimating the wheat grain yield (GY) based on selected soil parameters. These parameters, including the total electrical conductivity (EC), exchangeable sodium percentage (ESP), soil organic matter (OM), bulk density (BD), infiltration rate (IR), and sodium adsorption ratio (SAR), are considered pivotal in predicting the GY. The algorithm is trained and tested using a split of 70% for training and 30% for testing, as depicted in Figure 7. The KNN model employs regression to predict continuous variables, such as GY, by averaging the values of the k-nearest neighbors. In Figure 7A, the optimal target prediction for the soil quality index (SQI%) is identified as 7 t ha^−1^. This value serves as a reference point where the model assesses the grain yield to be favorable based on the specified predictors [101,102]. Subsequently, the model estimates the associated predictor values required to achieve the target GY, as illustrated in Figure 7B. The predicted GY is determined to be 7.257 t ha^−1^ under the following conditions: EC = 5.54 dS m^−1^, ESP = 10.02%, OM = 1.41%, BD = 1.30 g cm^−3^, IR = 1.15 cm h^−1^, and SAR = 7.80%. These values represent specific soil conditions that are deemed conducive by the model for attaining the desired wheat grain yield. For instance, lower levels of EC, ESP, BD, and SAR, along with higher values of Ks and OM, are associated with improved soil quality and are considered favorable for wheat cultivation. By incorporating these predictor variables, the KNN model provides valuable insights into the soil conditions necessary to optimize the wheat grain yield.

## 5. Conclusions

N-ZnO applied topically appears to be an effective strategy for reducing abiotic stresses, particularly for soils that are saline–sodic. It also appears to have a good effect on wheat plant growth, particularly when coupled with some inorganic and organic additives. The integration of gypsum, compost, and N-ZnO application ameliorated the soil quality via the soil physicochemical properties and was related to an improvement in the wheat productivity and nutrient uptake. This integration (G + C + Z_2_) resulted in a 14.81%, 40.60%, 35.10%, and 91.50% reduction in the EC, SAR, ESP, and RSE, respectively, and increase in the OM, total porosity, and infiltration rate of 22.22%, 7.30%, and 124.0%, respectively. Furthermore, the G + C + Z_2_ treatment exhibited the most significant increase in plant biomass, with the grain yield reaching 7.07 Mg ha^−1^, straw yield at 9.93 Mg ha^−1^, 1000-grain weight recorded as 57.03 Mg ha^−1^, and a plant height of 98.0 cm. The results indicated that the G + C + Z_2_ treatment increased both the nutrient content and wheat plant uptake. PCA demonstrated that increasing the grain yield correlated negatively with the ESP and EC and was positively correlated with the OM, infiltration rate (Ks), and plant nutrient content. Using a supervised machine learning approach, through the k-nearest neighbors (KNN) algorithm, by increasing the wheat grain yield as a target elucidated that by the EC, ESP, OM, BD, Ks, and SAR reaching 5.54 dS m^−1^, 10.02%, 1.41%, 1.30 g cm^−3^, 1.15 cm h^−1^, and 7.80%, it results in the prediction of increasing the wheat grain yield to reach 7.25 t ha^−1^. Overall, these findings underscore the potential of foliar N-ZnO application in conjunction with compost and gypsum amendments to enhance soil quality, boost wheat productivity, and optimize nutrient uptake, offering valuable insights for sustainable agricultural practices.

## Figures and Tables

**Figure 1 nanomaterials-14-01450-f001:**
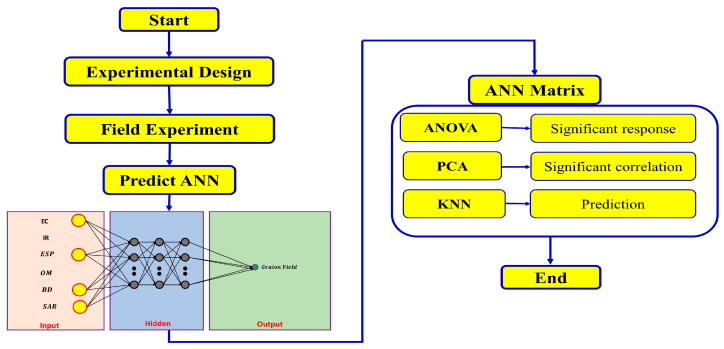
Experimental architecture diagram.

**Figure 2 nanomaterials-14-01450-f002:**
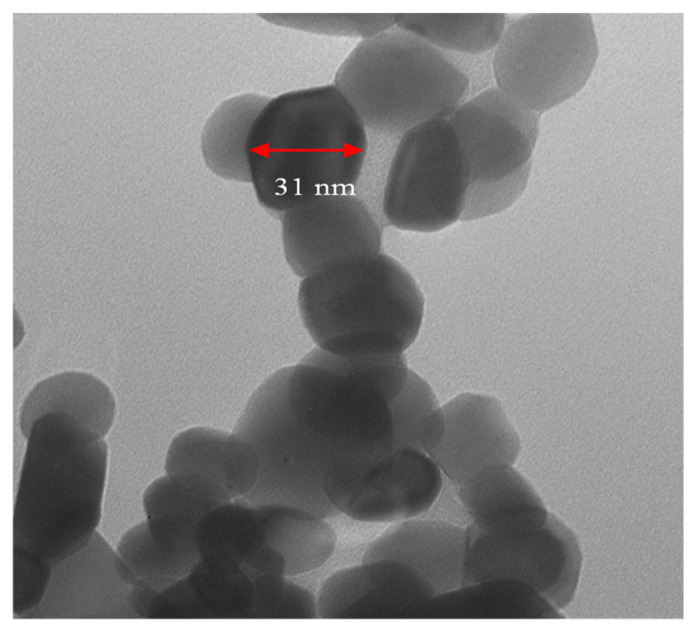
The TEM images of synthesized powder of N-ZnO.

**Figure 3 nanomaterials-14-01450-f003:**
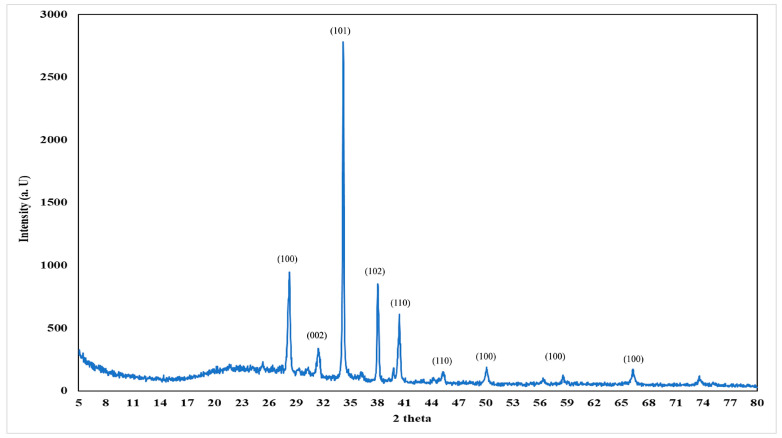
X-ray diffraction pattern of ZnO nano-powder.

**Figure 4 nanomaterials-14-01450-f004:**
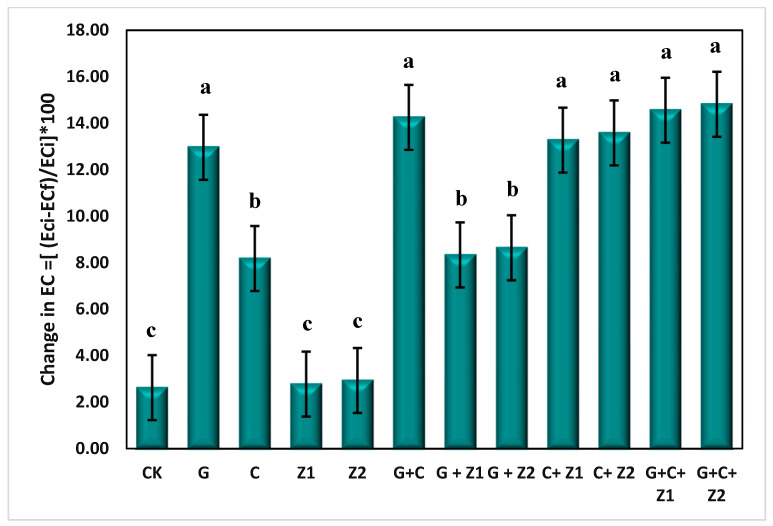
Changes in electrical conductivity (ECe) as affected by gypsum (G), compost (C), zinc foliar application, and its combination after wheat harvesting. Note that means with different letter are significantly different according to Duncan multiple range test (DMRT) at *p* < 0.05 at (*p* < 0.05) level, CK: control, G: gypsum, C: compost, Z_1_: zinc sulfate, Z_2_: N-ZnO.

**Figure 5 nanomaterials-14-01450-f005:**
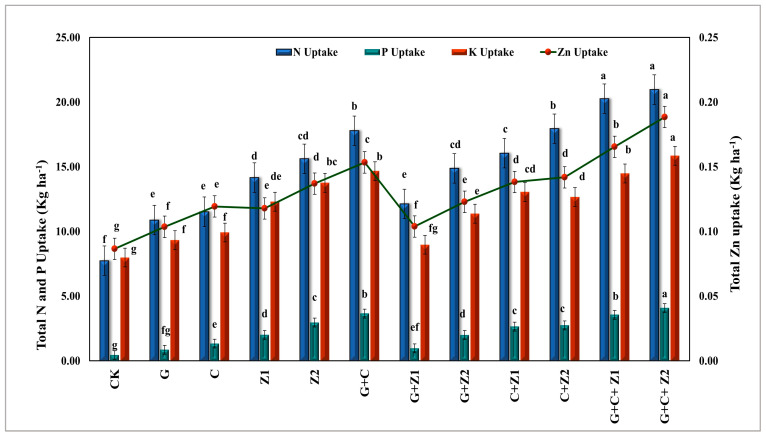
Effect of gypsum (G), compost (C), zinc foliar application, and its combination on total NPK and Zn uptake by wheat plants. Note that column in the same color with different letter are significantly different according to Duncan multiple range test (DMRT) at *p* < 0.05 at (*p* < 0.01) level, CK: control, G: gypsum, C: compost, Z_1_: zinc sulfate, Z_2_: N-ZnO.

**Figure 6 nanomaterials-14-01450-f006:**
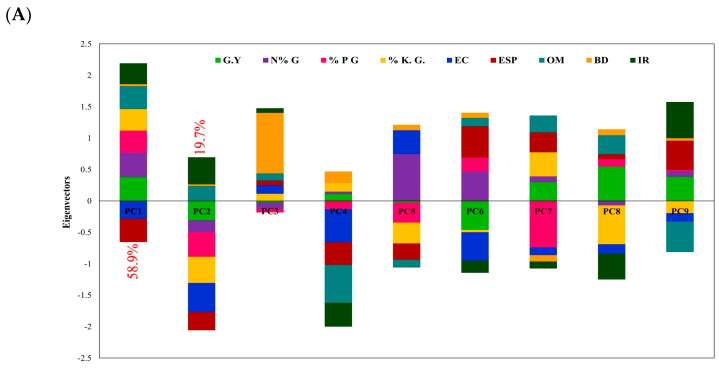
Correlations using (**A**) Eigenvectors components weights and (**B**) principal component analysis (PCA) of soil–plant properties as affected by gypsum (G), compost (C), zinc foliar application on wheat plant production.

**Figure 7 nanomaterials-14-01450-f007:**
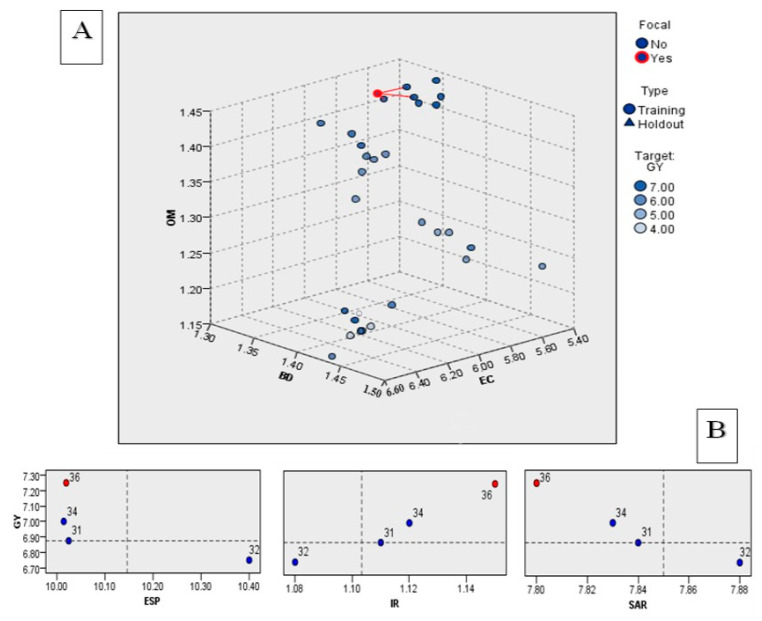
The k-nearest neighbors (KNN) model for grain yield as a target represent (**A**) predictor space dimensional between OM, BD, and EC using K neighbor = 3, and (**B**) quadrant map of different predictors including exchangeable sodium percent (ESP), infiltration rate (IR), and sodium adsorption ratio (SAR), and target grain yield value of 7 t ha^−1^.

**Table 1 nanomaterials-14-01450-t001:** Soil chemical and physical characteristics of the experimental site before cultivation.

Chemical Characteristics		Value	Physical Characteristics		Value
Soluble Ions, EC and pH			Particle Size Distribution (%)		
EC_e_ (dS m^−1^)		6.48	Clay		49.68
pH (soil suspension 1:2.5)		8.37	Silt		29.71
Soluble ions (mmol·L^−1^)			Sand		20.61
Na^+^		42.7	Texture class		Clayey
K^+^		2.6	Soil type		Saline–Sodic
Ca^++^		13.8	OM %		1.18
Mg^++^		5.7	Total CaCO_3_ (%)		2.28
Cl^−^		36.7	CEC (cmol_c_ kg^−1^)		38.20
HCO_3_^−^		22.9	Bulk density (g cm^−3^)		1.43
SO_4_^=^		5.2	Total porosity (%)		46.04
SAR (%)		13.69	Field capacity (%)		38.96
ESP %		16.05	Wilting point (%)		21.17
Available macronutrients (mg kg^−1^)					
N	31.76	P	8.16	K	305.7

**Table 2 nanomaterials-14-01450-t002:** Physicochemical properties of the rice straw compost.

Traits	Units	Values
pH	-	7.62
EC	dS m^−1^	4.12
N	%	2.53
P	%	1.23
K	%	1.41
Moisture ratio	-	17.80
B.D	g cm^−3^	0.57
O.M	%	55.78
Total-C	%	26.89
C/N Ratio	-	11.20

B.D: bulk density, Total-C: total organic carbon,

**Table 3 nanomaterials-14-01450-t003:** Effect of gypsum (G), compost (C), zinc foliar application, and its combination on saline–sodic chemical characteristics.

Treatments	ECe (dS m^−1^)	SAR (%)	ESP (%)	RSE (%)
CK	6.31 ± 0.01 ^a^	13.19 ± 0.01 ^a^	15.54 ± 0.05 ^a^	3.20 ± 0.01 ^i^
G	5.64 ± 0.04 ^c^	8.97 ± 0.02 ^d^	11.19 ± 0.01 ^d^	30.29 ± 0.01 ^e^
C	5.95 ± 0.02 ^b^	9.32 ± 0.02 ^c^	11.55 ± 0.01 ^c^	28.04 ± 0.01 ^g^
Z_1_	6.30 ± 0.30 ^a^	10.71 ± 0.01 ^b^	12.98 ± 0.01 ^b^	19.12 ± 0.01 ^h^
Z_2_	6.29 ± 0.01 ^a^	10.70 ± 0.01 ^b^	12.97 ± 0.01 ^b^	19.18 ± 0.01 ^h^
G + C	5.56 ± 0.06 ^c^	8.88 ± 0.01 ^f^	11.10 ± 0.10 ^d^	30.86 ± 0.01 ^b^
G + Z_1_	5.62 ± 0.02 ^c^	8.95 ± 0.04 ^de^	11.17 ± 0.01 ^d^	30.41 ± 0.01 ^d^
G + Z_2_	5.60 ± 0.10 ^c^	8.93 ± 0.03 ^e^	11.15 ± 0.03 ^d^	30.54 ± 0.05 ^c^
C + Z_1_	5.94 ± 0.04 ^b^	9.31 ± 0.01 ^c^	11.52 ± 0.02 ^c^	28.10 ± 0.10 ^g^
C + Z_2_	5.92 ± 0.02 ^b^	9.29 ± 0.01 ^c^	11.52 ± 0.02 ^c^	28.23 ± 0.03 ^f^
G + C + Z_1_	5.54 ± 0.04 ^c^	7.84 ± 0.04 ^g^	10.15 ± 0.22 ^e^	37.54 ± 0.05 ^a^
G + C + Z_2_	5.52 ± 0.02 ^c^	7.83 ± 0.03 ^g^	10.01 ± 0.01 ^e^	37.60 ± 0.010 ^a^
LSD (0.05)	0.16	0.04	0.12	0.08

SAR: sodium adsorption ratio, ESP: exchangeable sodium percentage, RSE: removal sodium efficiency. The column values with the same letters are statistical similar according to Duncan multiple range test (DMRT) at *p* < 0.05, LSD: least significant difference, CK: control, G: gypsum, C: compost, Z_1_: zinc sulfate, Z_2_: N-ZnO.

**Table 4 nanomaterials-14-01450-t004:** Effect of gypsum (G), compost (C), zinc foliar application, and its combination on soil physical characteristics.

Treatments	O.M (%)	B.D (g cm^−3^)	Total Porosity (%)	Infiltration Rate (cm h^−1^)
CK	1.17 ± 0.01 ^d^	1.42 ± 0.01 ^a^	46.42 ± 0.02 ^ef^	0.5 ± 0.1 ^c^
G	1.26± 0.01 ^c^	1.40 ± 0.10 ^abc^	47.17 ± 0.02 ^cde^	0.91 ± 0.01 ^b^
C	1.37 ± 0.02 ^b^	1.37 ± 0.01 ^abcd^	48.30 ± 0.20 ^abcd^	1.1 ± 0.1 ^a^
Z_1_	1.17 ± 0.01 ^d^	1.42 ± 0.02 ^a^	46.42 ± 0.02 ^ef^	0.5 ± 0.01 ^c^
Z_2_	1.18 ± 0.01 ^d^	1.41 ± 0.01 ^ab^	46.79 ± 0.01 ^def^	0.51 ± 0.01 ^c^
G+C	1.42 ± 0.01 ^a^	1.36 ± 0.02 ^abcd^	49.43 ± 0.03 ^a^	1.12 ± 0.02 ^a^
G + Z_1_	1.24 ± 0.01 ^c^	1.40 ± 0.02 ^abc^	45.17 ± 3.48 ^f^	0.91 ± 0.01 ^b^
G + Z_2_	1.25 ± 0.02 ^c^	1.39 ± 0.01 ^abcd^	47.68 ± 0.28 ^cde^	0.91 ± 0.01 ^b^
C+ Z_1_	1.39 ± 0.02 ^b^	1.36 ± 0.01 ^abcd^	48.68 ± 0.28 ^abc^	1.09 ± 0.01 ^a^
C+ Z_2_	1.39 ± 0.01 ^b^	1.35 ± 0.05 ^bcd^	49.06 ± 0.01 ^ab^	1.08 ± 0.02 ^a^
G + C+ Z_1_	1.42 ± 0.02 ^a^	1.34 ± 0.04 ^cd^	49.43 ± 0.04 ^a^	1.11 ± 0.03 ^a^
G + C+ Z_2_	1.43 ± 0.02 ^a^	1.33 ± 0.03 ^d^	49.81 ± 0.01 ^a^	1.12 ± 0.03 ^a^
LSD (0.05)	0.02	0.07	1.72	0.07

Column values with the same letters are statistical similar according to Duncan multiple range test (DMRT) at *p* < 0.05, LSD: least significant difference, CK: control, G: gypsum, C: compost, Z_1_: zinc sulfate, Z_2_: N-ZnO.

**Table 5 nanomaterials-14-01450-t005:** Average values of wheat plant biomass as affected by gypsum (G), compost (C), zinc foliar application, and its combination.

Treatments	Grain Yield (Mg ha^−1^)	Straw Yield (Mg ha^−1^)	1000 Grain Weight (Mg ha^−1^)	Harvest Index (%)	Plant Height (cm)
CK	3.85 ± 0.03 ^h^	7.77 ± 0.28 ^f^	32.34 ± 0.21 ^h^	33.12 ± 0.25 ^d^	63.33 ± 1.15 ^h^
G	5.25 ± 0.13 ^fg^	8.00 ± 0.25 ^def^	40.09 ± 0.95 ^fg^	39.64 ± 0.13 ^bc^	71.33 ± 2.52 ^g^
C	5.59 ± 0.03 ^ef^	8.33 ± 0.30 ^def^	42.67 ± 0.19 ^ef^	40.16 ± 0.29 ^bc^	77.33 ± 2.08 ^ef^
Z_1_	6.13 ± 0.25 ^cd^	8.38 ± 0.22 ^cde^	46.77 ± 1.91 ^cd^	42.24 ± 0.13 ^ab^	78.33 ± 2.31 ^ef^
Z_2_	6.67 ± 0.31 ^b^	8.62 ± 0.22 ^c^	50.91 ± 2.4 ^b^	43.61 ± 0.18 ^a^	80.33 ± 3.31 ^de^
G + C	6.92 ± 0.19 ^ab^	9.40 ± 0.02 ^ab^	52.82 ± 1.45 ^b^	42.40 ± 0.20 ^ab^	88.00 ± 0.00 ^bc^
G + Z_1_	4.96 ± 0.19 ^g^	7.91 ± 0.39 ^ef^	37.86 ± 1.46 ^g^	38.54 ± 0.24 ^c^	70.67 ± 1.15 ^g^
G + Z_2_	5.83 ± 0.31 ^de^	8.00 ± 0.29 ^def^	44.55 ± 2.40 ^de^	42.19 ± 0.03 ^ab^	74.33 ± 2.52 ^fg^
C + Z_1_	6.13 ± 0.25 ^cd^	8.56 ± 0.26 ^sd^	46.77 ± 1.91 ^cd^	41.70 ± 0.03 ^ab^	79.67 ± 2.52 ^de^
C + Z_2_	6.29 ± 0.31 ^c^	8.89 ± 0.64 ^bc^	48.05 ± 2.40 ^c^	41.48 ± 0.40 ^abc^	83.33 ± 5.86 ^cd^
G + C + Z_1_	6.83 ± 0.07 ^ab^	9.54 ± 0.31 ^a^	52.18 ± 0.55 ^b^	41.74 ± 0.25 ^ab^	89.00 ± 3.00 ^b^
G + C + Z_2_	7.07 ± 0.16 ^a^	9.93 ± 0.40 ^a^	57.03 ± 1.21 ^a^	41.58 ± 0.25 ^abc^	98.00 ± 3.00 ^a^
LSD	0.37	0.57	2.79	0.37	4.81

Column values with the same letters are statistical similar according to Duncan multiple range test (DMRT) at *p* < 0.05, LSD: least significant difference, CK: control, G: gypsum, C: compost, Z_1_: zinc sulfate, Z_2_: N-ZnO.

**Table 6 nanomaterials-14-01450-t006:** Effect of gypsum (G), compost (C), Zn foliar application, and its combination on nutrients contents in grain and straw of wheat plants.

Treatments	Grains				Straw			
	N (%)	P (%)	K (%)	Zn (mg kg^−1^)	N (%)	P (%)	K (%)	Zn (mg kg^−1^)
CK	0.98 ± 0.04 ^h^	0.07 ± 0.01 ^h^	0.30 ± 0.01 ^f^	9.57 ± 0.00 ^i^	0.51 ± 0.03 ^g^	0.02 ± 0.00 ^h^	0.89 ± 0.07 ^d^	3.14 ± 0.00 ^h^
G	1.16 ± 0.10 ^g^	0.10 ± 0.02 ^gh^	0.32 ± 0.03 ^e^	10.68 ± 0.10 ^h^	0.61 ± 0.03 ^efg^	0.04 ± 0.01 ^gh^	0.96 ± 0.07 ^d^	3.46 ± 0.00 ^gh^
C	1.24 ± 0.04 ^fg^	0.13 ± 0.05 ^fg^	0.34 ± 0.01 ^e^	11.90 ± 0.01 ^f^	0.55 ± 0.06 ^fg^	0.07 ± 0.01 ^f^	0.97 ± 0.05 ^d^	3.62 ± 0.01 ^fg^
Z_1_	1.37 ± 0.01 ^de^	0.19 ± 0.01 ^cd^	0.40 ± 0.01 ^cd^	11.22 ± 0.00 ^g^	0.69 ± 0.06 ^cde^	0.11 ± 0.01 ^e^	1.18 ± 0.03 ^bc^	3.90 ± 0.10 ^def^
Z_2_	1.49 ± 0.00 ^c^	0.22 ± 0.01 ^bc^	0.41 ± 0.01 ^abc^	12.64 ± 0.10 ^de^	0.66 ± 0.02 ^def^	0.17 ± 0.02 ^c^	1.28 ± 0.04 ^a^	4.21 ± 0.00 ^bcd^
G + C	1.54 ± 0.05 ^c^	0.24 ± 0.01 ^ab^	0.43 ± 0.01 ^ab^	13.03 ± 0.67 ^c^	0.76 ± 0.03 ^abcd^	0.21 ± 0.03 ^ab^	1.24 ± 0.09 ^ab^	4.49 ± 0.37 ^ab^
G + Z_1_	1.31 ± 0.01 ^ef^	0.09 ± 0.01 ^h^	0.32 ± 0.02 ^e^	10.76 ± 0.01 ^h^	0.71 ± 0.06 ^bcde^	0.06 ± 0.01 ^fg^	0.93 ± 0.02 ^d^	3.78 ± 0.00 ^efg^
G + Z_2_	1.45 ± 0.01 ^cd^	0.15 ± 0.01 ^ef^	0.38 ± 0.02 ^d^	12.43 ± 0.01 ^e^	0.80 ± 0.02 ^abc^	0.14 ± 0.02 ^d^	1.15 ± 0.05 ^c^	4.04 ± 0.00 ^cde^
C + Z_1_	1.47 ± 0.08 ^cd^	0.17 ± 0.01 ^de^	0.41 ± 0.01 ^bc^	13.10 ± 0.10 ^c^	0.83 ± 0.07 ^ab^	0.19 ± 0.01 ^bc^	1.23 ± 0.02 ^abc^	4.26 ± 0.03 ^bc^
C + Z_2_	1.66 ± 0.06 ^b^	0.21 ± 0.04 ^bc^	0.39 ± 0.02 ^cd^	12.87 ± 0.01 ^cd^	0.85 ± 0.11 ^a^	0.16 ± 0.00 ^cd^	1.15 ± 0.04 ^c^	4.38 ± 0.54 ^bc^
G + C+ Z_1_	1.78 ± 0.15 ^a^	0.24 ± 0.02 ^ab^	0.41 ± 0.01 ^bc^	14.18 ± 0.01 ^b^	0.84 ± 0.17 ^a^	0.20 ± 0.01 ^ab^	1.23 ± 0.05 ^abc^	4.42 ± 0.00 ^b^
G + C+ Z_2_	1.81 ± 0.00 ^a^	0.25 ± 0.02 ^a^	0.43 ± 0.01 ^a^	15.55 ± 0.01 ^a^	0.82 ± 0.07 ^ab^	0.23 ± 0.01 ^a^	1.29 ± 0.06 ^a^	4.81 ± 0.01 ^a^
LSD (0.05)	0.11	0.03	0.03	0.34	0.13	0.03	0.09	0.34

Column values with the same letters are statistical similar according to Duncan multiple range test (DMRT) at *p* < 0.05, LSD: least significant difference, CK: control, G: gypsum, C: compost, Z_1_: zinc sulfate, Z_2_: N-ZnO.

## Data Availability

The datasets used and analyzed during the current study are available from the corresponding author on reasonable request.

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
