# Peer review of "Effect of Nano-Zinc Oxide, Rice Straw Compost, and Gypsum on Wheat (Triticum aestivum L.) Yield and Soil Quality in Saline–Sodic Soil"

_nanomaterials, 2024, doi:10.3390/nano14171450_

Round 1

Reviewer 1 Report

Comments and Suggestions for Authors

Block diagrams are very useful, and their use would be highly recommended in the materials and methods section. This would prevent the reader from getting confused and would make the procedures very clear.

The brand, model and analysis modes of all the equipment used must be written, especially the transmission electron microscope and the X-ray diffractometer.

The methods used to calculate porosity should be written down, indicating the type of porosity (open or closed). Ideally, a high-resolution Scanning Electron Microscopy photomicrograph should be included to observe all the topographic details of the samples. If analysis is not possible, at least include an image of the initial condition, as well as the best composition.

The phrase unitary operations and processing must be integrated in the paragraph on lines 188 to 189.

In lines 200 to 201 please add that the samples of… were analyzed spectrophotometrically using the atomic absorption chemical analysis method.

The methods by which the porosity was calculated should be written and the type of porosity should be indicated (open or closed). Ideally, a high-resolution Scanning Electron Microscopy photomicrograph should be included to observe all the topographic details of the samples. If the analysis is not possible, at least include an image of the initial condition, as well as the best composition.

It is very important to add to the text of the article-tables what percentage it is, whether by weight or by volume.

Also, in general, when writing about the specific surface area, it is advisable to write a numerical value, for example, what the area was before and after processing in the samples.

It is very important to write if the adsorption calculations were based on a mathematical model.

It is advisable not to use apostrophes anywhere in the manuscript, such as in lines: 433, 458 and 465.

It is very important to write numerical values ​​from the reviewed literature when related to macro and microporosity. For example, in line 499.

Transmission Electron Microscopy photomicrographs must be edited and marked which areas could be attributed to organic and/or inorganic material.

Comments on the Quality of English Language

No comments.

Author Response

Thanks for revision this manuscript and for your kind words. We have considered all your comments.

Comments to the Author

1 - Block diagrams are very useful, and their use would be highly recommended in the materials and methods section. This would prevent the reader from getting confused and would make the procedures very clear.

Thanks for your suggestions. We have provided a block diagram for the treatments for more clarification and added an architecture building diagram for the experimental procedure according to your suggestion.

2 - The brand, model and analysis modes of all the equipment used must be written, especially the transmission electron microscope and the X-ray diffractometer.

The details of the equipment’s have been added.

3- The methods used to calculate porosity should be written down, indicating the type of porosity (open or closed). Ideally, a high-resolution Scanning Electron Microscopy photomicrograph should be included to observe all the topographic details of the samples. If analysis is not possible, at least include an image of the initial condition, as well as the best composition. 

The method of calculation the total porosity has been added (Lines: 202-205). Unfortunately, because of low budget we could not afford the SEM photomicrograph for both initial and later conditions, but we instead used TEM image regarding those details.

4- nThe phrase unitary operations and processing must be integrated in the paragraph on lines 188 to 189.

The phrase has been modified (Lines 197-198).

5-  In lines 200 to 201 please add that the samples of… were analyzed spectrophotometrically using the atomic absorption chemical analysis method.

It has been added.

 6- It is very important to add to the text of the article-tables what percentage it is, whether by weight or by volume.

As we find this comment is hard to understand, we hypothesis that you ask about the percentage of each variable in the manuscript. All percentages in the manuscript is based on concentrations of elements as mentioned in the text for SAR, ESP, RSE (%).

7- Also, in general, when writing about the specific surface area, it is advisable to write a numerical value, for example, what the area was before and after processing in the samples.

We add in line 497 and 498 surface area of nano-ZnO in this study =73.10 m2 g-1)

8- It is very important to write if the adsorption calculations were based on a mathematical model.  The adsorption has mentioned just for Sodium adsorption ratio (SAR) in the manuscript. Its calculation based on the equation (2) presented at Line 218.

9-It is advisable not to use apostrophes anywhere in the manuscript, such as in lines: 433, 458 and 465.

You are right. It has been modified.

10- It is very important to write numerical values ​​from the reviewed literature when related to macro and microporosity. For example, in line 499.

The literature modified (Lines 538-541).

11- Transmission Electron Microscopy photomicrographs must be edited and marked which areas could be attributed to organic and/or inorganic material.

Thanks for your suggestion. However, as the synthesize of the Nano-ZnO were chemical procedure, there were no organic materials needs to be highlighted.

Reviewer 2 Report

Comments and Suggestions for Authors

The paper by El Sharkawy et al provides valuable information about the importance of adapting cropping conditions of wheat to harsh environmental changes. The workflow is good and the experimental setup provides sufficient information for other large-scale experiments. However, some information needs readjustments in the manuscript:

1.       The title. I believe that statistical analysis is a mandatory component in experimental setups, therefore it is not necessary to be present in the title. With few adjustments, the title can be improved.

2.       Line 134. I suppose it should be “located” instead of “lactated”

3.       Table 2 caption: without “Some”

4.       Line 155: I suggest mentioning the season (month) as well, along with the information already provided

5.       Line 181: please find another term instead of “toasted” (incubated, heated, etc.)

6.       The EC abbreviation is somewhat confusing. In the abstract you say “soil salinity (as EC)” and then it appears in Table 2 again and then it is defined as electrical conductivity in line 190. Try to better clarify this abbreviation and the rest if that is the case.

7.       Describe the weather conditions in M&M (temperature, humidity, precipitations, sun light, etc.) for the entire experiment.

8.       Line 240, TEM is in figure 1, not 2

9.       Line 246, the nanoparticles are polygonal and not rods. The percentage of rods from the images you provided is small and it cannot be taken into consideration as the main morphology for the nanoparticles.

10.   Line 251, Figure 2, not Fig 1

11.   Line 259, I suggest moving the equation in M&M

12.   Fig 1, I suggest keeping only one TEM micrograph, since both say the same information

13.   Figure 3. You have statistical analyses performed for this graph. State in the figure caption what statistical was made, p values, compared to which control group, and what do the letters stand for (as you did for Table 3). Please do the same for all figures and tables.

14.   Table 3, lines 314-316 repeat; also, describe all abbreviation underneath the table (CK for example)

15.   Line 351, Figure 4?

16.   Figure 4 needs improvement. What are the red bars? Add legend for the line as well and try to find a way to distinguish the statistics from the line from that from the bars (same color code or something like that)

17.   I think that figures 5 and 6 should be moved in the results section

18.   The PCA. I might be wrong, but I think the PCA is incomplete and not well described in the text. What did you use as PC1 and PC2? you have multiple test samples, did you compare all of them with the control group, or did you compare within them? Add more details to the text, move it to the results section, after Table 6 or Figure 4.

Author Response

Thanks for revision this manuscript and for your kind words. We have considered all your comments.

Comments to the Author

  • The title. I believe that statistical analysis is a mandatory component in experimental setups, therefore it is not necessary to be present in the title. With few adjustments, the title can be improved.
  • The title has been modified, but it is up to the editor to accept that change.
  • Line 134. I suppose it should be “located” instead of “lactated”

- You are right. It has been corrected (Line 133).

  • Table 2 caption: without “Some”
  •  
  • Line 155: I suggest mentioning the season (month) as well, along with the information already provided.
  • Suggestion added (Lines 159-160).
  • Line 181: please find another term instead of “toasted” (incubated, heated, etc.)

- You are right. That word was added just for plagiarism and has been changed to (heated).

  • The EC abbreviation is somewhat confusing. In the abstract you say “soil salinity (as EC)” and then it appears in Table 2 again and then it is defined as electrical conductivity in line 190. Try to better clarify this abbreviation and the rest if that is the case.
  • The Expression has been maintained and unified.
  • Describe the weather conditions in M&M (temperature, humidity, precipitations, sun light, etc.) for the entire experiment.
  • The weather conditions have been applied (Lines 135-137).
  • Line 240, TEM is in figure 1, not 2.
  • Figures have been maintained.
  • Line 246, the nanoparticles are polygonal and not rods. The percentage of rods from the images you provided is small and it cannot be taken into consideration as the main morphology for the nanoparticles.
  • We will take your suggestion and modify the shape of the nanoparticles.
  • Line 251, Figure 2, not Fig 1.
  • All Figures have been maintained.
  • Line 259, I suggest moving the equation in M&M.
  • Equation moved.
  • Fig 1, I suggest keeping only one TEM micrograph, since both say the same information.
  • One TEM image has been kept.
  • Figure 3. You have statistical analyses performed for this graph. State in the figure caption what statistical was made, p values, compared to which control group, and what do the letters stand for (as you did for Table 3). Please do the same for all figures and tables.
  • You are right. All figures’ captions have been added.
  • Table 3, lines 314-316 repeat; also, describe all abbreviation underneath the table (CK for example)
  • You are right. All Tables’ captions have been added.
  • Line 351, Figure 4?
  • Figures have been maintained.
  • Figure 4 needs improvement. What are the red bars? Add legend for the line as well and try to find a way to distinguish the statistics from the line from that from the bars (same color code or something like that)
  • You are right. Figure 4 has been modified.
  • I think that figures 5 and 6 should be moved in the results section
  • The PCA figure has been moved to the “Results” section. While, we keep the last Figure in Discussion part for more clarification of the core of the manuscript.
  • The PCA. I might be wrong, but I think the PCA is incomplete and not well described in the text. What did you use as PC1 and PC2? you have multiple test samples, did you compare all of them with the control group, or did you compare within them? Add more details to the text, move it to the results section, after Table 6 or Figure 4.
  • For more clarification, the PCA figure has been modified adding the details and extra sub-figure for Eigenvalues components.

thank you

Round 2

Reviewer 2 Report

Comments and Suggestions for Authors

Dear authors, thank you for revising the paper accordingly. I recommend the publication after several minor revisions. Congratulations on the results!

1.       Scherrer's formula appears twice in the text (see line 274)

2.       Please do not forget to mention which statistical analysis was made for each figure and table where applicable (ANOVA or DMRT)

3.       Uniformization is needed in Figure 6 considering the text in the figure (G.Y. or G Y; N% P, etc.)

Author Response

Response to Reviewers

Dear Prof. Dr. Editor of Nanomaterials Journal,

Following your remarks, we decided to revise our paper.

The changes are listed below in blue color words.

Manuscript ID " nanomaterials-3178251" entitled “Effect of nano-zinc oxide, rice straw compost and gypsum on wheat (Triticum aestivum L.) yield and soil quality in saline-sodic soil: K-Nearest Neighbors and Principal component analysis (PCA) techniques "

Reviewer(s)' Comments to Author:

Reviewer # 1:

Dear authors, thank you for revising the paper accordingly. I recommend the publication after several minor revisions. Congratulations on the results!

Thanks for revision this manuscript and for your kind words. We have considered all your comments.

Comments to the Author

- Scherrer's formula appears twice in the text (see line 274).

-   Corrected.

  • Please do not forget to mention which statistical analysis was made for each figure and table where applicable (ANOVA or DMRT).

-  The details have been added.

  • Uniformization is needed in Figure 6 considering the text in the figure (G.Y. or G Y; N% P, etc.).
  • It has been uniformed.
  • thank you
